
# Combined Impacts of Nitrous Acid and Nitryl Chloride on Lower Tropospheric Ozone: New Module Development in WRF-Chem and Application to China

Li Zhang[1], Qinyi Li[1], Tao Wang[1,*], Ravan Ahmadov[2,3], Qiang Zhang[4], Meng Li[4], and Mengyao Lv[5]

[1] Department of Civil and Environmental Engineering, The Hong Kong Polytechnic University, Hong Kong, China,

[2] Cooperative Institute for Research in Environmental Sciences, University of Colorado at Boulder, Boulder, CO, USA,

[3] Earth System Research Laboratory, National Oceanic and Atmospheric Administration, Boulder, CO, USA,

[4] Department of Earth System Science, Tsinghua University, Beijing, China,

[5] National Meteorological Center, China Meteorological Administration, Beijing, China

*Correspondence to:* T. Wang (cetwang@polyu.edu.hk)

**Abstract.** Nitrous acid (HONO) and nitryl chloride ($ClNO_2$)—through their photolysis—can have profound effects on the nitrogen cycle and oxidation capacity of the lower troposphere. Previous numerical studies have separately considered and

investigated the sources/processes of these compounds and their roles in the fate of reactive nitrogen and the production of ozone ($O_3$), but their combined impact on the chemistry of the lower part of the troposphere has not been addressed yet. In this study, we updated the WRF-Chem model with the currently known sources and chemistry of HONO and chlorine in a new chemical mechanism (CBMZ_ReNOM), and applied it to a study of the combined effects of HONO and $ClNO_2$ on summertime $O_3$ in the boundary layer over China. We simulated the spatial distributions of HONO, $ClNO_2$, and related

compounds at the surface and within the lower troposphere. The results showed that the modeled HONO levels reached up to 800–1800 ppt at the surface (0–30 m) over the Northern China Plain (NCP), the Yangtze River Delta (YRD), and the Pearl River Delta (PRD) regions and that HONO was concentrated within a 0–200 m layer. In comparison, the simulated surface $ClNO_2$ mixing ratio was around 800–1500 ppt over the NCP, YRD, and central China regions and was predominantly present in a 0–600 m layer. HONO enhanced daytime $RO_x$ (OH+$HO_2$+$RO_2$) and $O_3$ at the surface (0–30 m) by 2.8–4.6 ppt

(28–37%) and 2.9–6.2 ppb (6–13%), respectively, over the three most developed regions, whereas $ClNO_2$ increased surface $O_3$ in the NCP and YRD regions by 2.4–3.3 ppb (or 5–6%) and it also had a significant impact (3–6%) on above-surface $O_3$ within 200–500 m. The combined effects increased surface $O_3$ by 11.5%, 13.5%, and 13.3% in the NCP, YRD and PRD regions, respectively. Over the boundary layer (0–1000m), the HONO and $ClNO_2$ enhanced $O_3$ by up to 5.1% and 3.2%, respectively, and their combined effect increased $O_3$ by 7.1–8.9% in the three regions. The new module noticeably improved

$O_3$ predictions at ~900 monitoring stations throughout China by reducing the mean bias from -4.3 ppb to 0.1 ppb. Our study suggests the importance of considering these reactive nitrogen species simultaneously into chemical transport models to better simulate the formation of summertime $O_3$ in polluted regions.


## 1 Introduction

Reactive nitrogen compounds play important roles in atmospheric chemistry and affect the formation of secondary pollutants including ozone ($O_3$) and secondary aerosols. In recent decades, nitrous acid (HONO), dinitrogen pentoxide ($N_2O_5$), and nitryl chloride ($ClNO_2$) have received considerable attention, together with the traditionally known oxides of nitrogen

($NO+NO_2$), due to their potentially significant impact on the oxidation capacity in the polluted portion of the troposphere (Brown, 2006; Su et al., 2011; Thornton et al., 2010). HONO can be heterogeneously formed via conversions of $NO_2$ (R1) on particle, land, and sea surfaces (Kleffmann, 2007; Zha et al., 2014), directly emitted by traffic and biological activities (Kurtenbach et al., 2001; Oswald et al., 2013), and formed via photolysis of nitric acid (Zhou et al., 2011). During the day, HONO is then photochemically converted into OH radicals and NO (R2), both of which have an impact on the oxidation

capacity (Kleffmann, 2007). $N_2O_5$, a night reservoir of $NO_x$, can react on particles containing chloride to form $ClNO_2$ (R3). Within a few hours after sunrise, the $ClNO_2$ is photolyzed into $NO_2$ and reactive chlorine atoms (R4), both of which react with hydrocarbons to produce additional peroxy radicals ($RO_2$ and $HO_2$) and subsequently accelerate the cycling of oxidants in the atmosphere at both ground level and upper levels (Tham et al., 2016). The photolysis of HONO and $ClNO_2$ during the daytime could have a significant effect on the formation of secondary pollutants in polluted regions (Osthoff et al., 2008;

Tham et al., 2016; Tang et al., 2015; Thornton et al., 2010).

$$2NO_{2(g)} + H_2O \rightarrow HONO_{(g)} + HNO_{3(g)} \tag{R1}$$

$$HONO_{(g)} + hv \rightarrow OH^* + NO_{(g)} \tag{R2}$$

$$N_2O_{5(g)} + Cl^-_{(aq)} \rightarrow ClNO_{2(g)} + NO_3^-_{(aq)} \tag{R3}$$

$$ClNO_{2(g)} + hv \rightarrow Cl^*_{(g)} + NO_{2(g)} \tag{R4}$$

Previous numerical studies have separately considered the sources/processes of HONO and $ClNO_2$-initiated chlorine chemistry and investigated their respective effects on $O_3$ in different regions. By considering direct and secondary HONO sources, a few studies using regional chemical transport models (e.g. CMAQ, WRF-Chem) suggested that HONO could noticeably enhance surface-level $O_3$ by 1.4–6 ppb in North America (Li et al., 2010; Sarwar et al., 2008), 1–3 ppb in Europe (Elshorbany et al., 2012; Gonçalves et al., 2012), and 3–12 ppb in Northern China (Li et al., 2011; An et al., 2013). A very

recent study by our group parameterized up-to-date HONO sources, including gas-phase reactions, heterogeneous formation at various surfaces, traffic sources, and biological emissions, into a chemistry transport model (WRF-Chem). The improved model well reproduced the HONO observed at a suburban site with complex terrain in Hong Kong and indicated an enhancement of 10–12% in ground-level $O_3$ concentration over the Pearl River Delta region (PRD), China (Zhang et al., 2016).





Several chemical transport model studies have evaluated the impact of the hydrolysis of $N_2O_5$, the subsequent formation of $ClNO_2$, and/or Cl-radical-initiated chemistry on tropospheric chemistry in North America and Europe. These studies suggested that these new chemistry could remarkably influence NOx and chlorine cycling, leading to an enhancement of 1–2 ppb in surface $O_3$ concentration in Texas (Simon et al., 2010), a 3–4% increase in monthly average 8-h $O_3$ across the U.S. (Sarwar et al., 2012), and a negative impact on nighttime oxidative chemistry with reductions of up to 30% in $NO_3$ and $N_2O_5$ (without considering chlorine chemistry) over the north-western Europe (Archer-Nicholls et al., 2014; Lowe et al., 2015). In a recent study, we parameterized the heterogeneous uptake of $N_2O_5$, the formations of $ClNO_2$, and chlorine chemistry into WRF-Chem, and demonstrated that these processes could lead to $O_3$ enhancement of up to 16.3% within the planetary boundary layer (PBL) over the PRD region in the winter season (Li et al., 2016).

Despite the above-mentioned research on the effects of HONO and $N_2O_5$/$ClNO_2$, a comprehensive assessment of their combined impacts on $O_3$ is lacking. The sources (and thus the impacts) of HONO are mostly near the ground surface (Su et al., 2011; Zhang et al., 2016), whereas significant amounts of $ClNO_2$ form within the residual layer, followed by downward mixing after breakup of the nocturnal inversion layer in the morning (Tham et al., 2016; Wang et al., 2016). Moreover, the chemical processes of HONO and $N_2O_5$/$ClNO_2$ occur concurrently and are coupled in the troposphere, both of which would simultaneously shift the composition of the total reactive nitrogen ($NO_y$), influence the nitrogen oxide chemistry, and affect the formations of secondary pollutants. To the best of our knowledge, no global or regional models, however, have simultaneously considered the sources/processes of HONO and $ClNO_2$ and evaluated their regional impacts on the formation of $O_3$ pollution in the boundary layer of the atmosphere.

In the present study, we developed a new chemical mechanism option in WRF-Chem to consider both the detailed HONO sources/processes and chlorine chemistry and applied the new mechanism to investigate their respective and combined effects on summertime $O_3$ pollution in the lower troposphere over China. In the remainder of this paper we describe the new developments in WRF-Chem, detailed model configurations, and measurement data used in the study in section 2. Then, in section 3, we show the model simulations of HONO, $N_2O_5$, and $ClNO_2$ over China during a 10-day period in summer and compare the results against available measurements. Following this, we illustrate the impacts of HONO and $ClNO_2$ on $O_3$ pollution in China. The major findings of the study are given in section 4.

## 2 Methodology

### 2.1 Development of CBMZ_ReNOM

A new chemical mechanism option in WRF-Chem, namely CBMZ_ReNOM, was further developed based on the CBMZ mechanism to include detailed HONO chemistry and chlorine chemistry in this study. The CBMZ gas-phase chemical mechanism was developed by Zaveri and Peters (1999). It treats 67 species and 164 reactions using a lumped-structure





approach that categorizes organic compounds according to the types of bonds present in their molecular structures. The original mechanism does not contain any direct emissions or secondary formation of HONO except for the well-known homogeneous formation via OH and NO, nor does it consider any chemical reactions involving chlorine-containing species. We incorporated the following new chemistry into the CBMZ_ReNOM module based on the CBMZ mechanism.

### 2.1.1 HONO chemistry

In our previous study, we expanded the original CBMZ mechanism by adding up-to-date comprehensive sources of HONO, including gas-phase formations (in addition to the reaction between OH and NO), heterogeneous formations, direct traffic sources, and soil bacteria emissions as described by Zhang et al. (2016). We showed that including these additional sources of HONO reproduced the observed HONO by 85% on average at a suburban site in southern China. In this study, we added similar HONO-related processes, which are briefly summarized below.

**Heterogeneous formations**

$$NO + HNO_3 \xrightarrow{surface} HONO + NO_2 \tag{R5}$$

$$NO + NO_2 + H_2O \xrightarrow{surface} 2HONO \tag{R6}$$

$$2HONO \xrightarrow{surface} NO + NO_2 + H_2O \tag{R7}$$

$$2NO_2 + H_2O \xrightarrow{surface} HONO + HNO_3 \tag{R8}$$

All of the heterogeneous reactions were considered as simple first-order reactions and the formed HONO was assumed to out-gas instantaneously. The reaction rates for R5–R7 were obtained from Foley et al. (2010), and the first-order reaction rate for R8 was estimated following the recommendations of Zhang et al. (2016) for different surfaces.

**Direct emissions**

The model also considers HONO directly emitted from anthropogenic sources (vehicles and water vessels) and biological activities. A widely used traffic emission ratio of 1.6% (ratio of HONO to $NO_2$ in the traffic emission sector) was applied to parameterize traffic emission of HONO. For direct emissions from soil bacteria, we followed the parameterization of Zhang et al. (2016) with consideration of the dependence on land category, soil humidity, and temperature. In brief, we first mapped the optimum HONO emission fluxes of various soil categories (from 17 ecosystems) measured by Oswald et al. (2013) into the most closely matching United States Geological Survey (USGS) land categories in the WRF-Chem model following the mapping schemes described by Zhang et al. (2016). Then, the emission flux for each USGS land-use type was calculated as the aggregation of the measured fluxes from the measured category/categories that was/were mapped into the specific USGS





classifications. After that, the optimum fluxes over the domain were incorporated into the model and were further scaled online according to the soil temperature and water content in each model grid at each time step throughout the simulation period. More details of the parameterization can be found in Zhang et al. (2016).

### 2.1.2 Heterogeneous $N_2O_5$ chemistry and chlorine chemistry

For the heterogeneous formation of $ClNO_2$ through the reactions of $N_2O_5$ on Cl-containing particles, we applied the modified MOSAIC module developed by Archer-Nicholls et al. (2014) and Lowe et al. (2015). The heterogeneous processes of $N_2O_5$ were considered as simple first-order reactions, taking into account the particle surface area density, molecular velocity, and uptake coefficient of $N_2O_5$ on an aerosol surface. Parametrization of the uptake coefficient of $N_2O_5$ followed the method reported by Bertram and Thornton (2009) and further considered its suppression by organic coatings using the method

proposed by Riemer et al. (2009). More details on the modifications were introduced by Archer-Nicholls et al. (2014) and Lowe et al. (2015).

In the treatment of heterogeneous $N_2O_5$ chemistry in WRF-Chem by Archer-Nicholls et al. (2014), $ClNO_2$ was assumed to out-gas near-instantaneously, and it was treated as an inert species without further gas-phase reactions. Based on their development, we further included the chlorine chemical reactions listed in Table 1 into the CBMZ_ReNOM module. The

chlorine mechanism introduced in the present study further extends the one proposed by Sarwar et al. (2012), which was based on CB05, and includes more chlorine species and reactions. It consists of six photolysis reactions (photolysis of $Cl_2$, HOCl, $ClNO_2$, $ClONO_2$, and FMCl) and 30 gas-phase reactions (R7–R36 as listed in Table 1), and introduces seven new chemical species (Cl, $Cl_2$, ClO, HOCl, $ClNO_2$, $ClONO_2$, and FMCl) into the original CBMZ mechanism. The reactions between lumped volatile organic compounds (VOCs) species and Cl radicals are similar to their reactions with OH radicals

(Table 1).

### 2.1.3 Photolysis module

The photolysis rates of the newly added chlorine-containing chemicals, i.e., $Cl_2$, HOCl, $ClNO_2$, $ClONO_2$, and FMCl (reactions 1–6 in Table 1), in the CBMZ_ReNOM module were considered in both the Fast-J and Madronich photolysis modules in WRF-Chem. The photolysis rates were calculated using Eq. (1), which depends on the quantum yield ($\phi$) and

absorption cross-section ($\sigma$) of each chemical species ($i$), as functions of wavelength and temperature, and on the actinic flux (F), as a function of solar radiation and wavelength.

$$J_i = \int \phi_i(\lambda, T)\sigma_i(\lambda, T)F(\lambda)d\lambda \tag{1}$$

The absorption cross sections and quantum yields for the reactions were obtained from the atmospheric chemical kinetic database of the International Union of Pure and Applied Chemistry (http://iupac.pole-ether.fr/) and were interpolated into the

wavelengths that used in the Fast-J and Madronich modules, while the actinic flux was calculated in the photolysis modules





(Wild et al., 2000; Madronich, 1987). Only the modified Fast-J module was applied for providing the photolysis rates in the present study.

## 2.2 WRF-Chem model set up

### 2.2.1 Model configurations

The three-dimensional Weather Research and Forecasting coupled with Chemistry (WRF-Chem) model (https://ruc.noaa.gov/wrf/wrf-chem/) simulates the transport, mixing, and chemical transformation of trace gases and aerosols simultaneously with meteorology (Grell et al., 2005). The chemical mechanism used to simulate the gases and aerosols is based on the CBMZ/CBMZ_ReNOM module coupled with the sectional Model for Simulating Aerosol Interactions and Chemistry (MOSAIC, Zaveri et al., 2008) with new parameterizations of the heterogeneous uptake of $N_2O_5$

and $ClNO_2$ formation (Archer-Nicholls et al., 2014). Other major physical and chemical schemes applied in our WRF-Chem simulations included the Goddard shortwave radiation scheme (Chou et al., 1998), the rapid radiative transfer model (RRTM) long-wave radiation scheme (Mlawer et al., 1997a), the Mellor-Yamada-Janjic (MYJ) PBL scheme (Janjić, 1994), the modified double-moment version of the Lin microphysics scheme (Lin et al., 1983), and the Grell-Dévényi cumulus scheme (Grell and Dévényi, 2002) (see Table 2). The simulation domain in this study was designed to cover most of China

with a resolution of $27 \times 27$ km as illustrated in Figure 1. The vertical resolution included 31 layers with a fixed-model top pressure of 100 hPa, with the first model layer set to be about 30 m above ground level (a.g.l.) and eight model layers below 1000 m a.g.l. (approximately the height of PBL at noon). Simulations by the Model for Ozone and Related Chemical Tracers (version 4) driven by Goddard Earth Observing System-5 fields were used to provide the initial and boundary conditions for WRF-Chem (Emmons et al., 2010).

Four simulation cases were designed, as listed in Table 3. In the base case, the default WRF-Chem model was employed. The ReNOM_HONO and ReNOM_Cl cases applied the new CBMZ_ReNOM module considering HONO chemistry and Cl-chemistry individually, respectively, whereas in the ReNOM case, the CBMZ_ReNOM module with both HONO and Cl-initiated reactions was used. A period during the summer season (26 June-7 July 2014) was selected for model simulations because ozone pollution is severe in the summer in most regions of China (Wang et al., 2017). The first 24 h of the

simulation were considered as a spin-up time. The four-dimensional nudging-based data assimilation method was used in our WRF-Chem simulations throughout the simulation period. This method effectively improves the meteorological performance of WRF-Chem (Zhang et al., 2015; Zhang et al., 2016). Meteorological observations at more than 3614 surface stations (tri-hourly) and 297 sounding stations (12-hourly) were obtained from the China Meteorological Administration and integrated into the simulations through observational nudging (Figure 1a). National Centers for Environmental Prediction

final reanalysis data (https://rda.ucar.edu/datasets/ds083.2/) were used in the analytical nudging.



### 2.2.2 Emission data

Three sets of anthropogenic emission inventories were used in our simulations. For mainland China, we applied the latest Multi-resolution Emission Inventory for China (MEIC) in 2013 with a resolution of $0.25° \times 0.25°$. The MEIC provides the monthly emissions of ten primary anthropogenic pollutants, including $SO_2$, $NO_x$, CO, $CO_2$, $NH_3$, $PM_{2.5}$, $PM_{10}$, BC, OC, and

nonmethane volatile organic compounds (NMVOCs), from the five major sectors of agriculture, industry, power plants, residential, and transportation (http://www.meicmodel.org). The emission inventory was developed by Tsinghua University based on a technology-based emission model (Lei et al., 2011; Zhang et al., 2009), and has been proved to offer reasonable model predictions of $PM_{2.5}$ and $O_3$ in multiple cities over China during multiple years (Zhang et al., 2015; Zhang et al., 2016). For other Asian regions, the emission inventory for Asia (MIX) in 2010 with a resolution of $0.25° \times 0.25°$ was used in

the simulations. The MIX emission inventory was developed for the Model Inter-Comparison Study for Asia (MICS-Asia) and covers all major anthropogenic sources in 30 Asian countries and regions (Li et al., 2017). More details of the MIX emission inventory can be found in (Li et al., 2017), and monthly emission datasets are available at http://www.meicmodel.org/dataset-mix. For chlorine emissions, the Reactive Chlorine Emission Inventory (RCEI; Keene et al. (1999) and references therein) was adopted. The RCEI, on a of $1° \times 1°$ grid scale, contains emissions of a total of nine

reactive chlorine species from both biomass burning and anthropogenic activities. Although the RCEI might be subject to large uncertainties in terms of representing Cl emissions in China due to its low spatial resolution and obsolete surrogate data (in 1990), as discussed in Li et al. (2016), we still applied this emission inventory because it is the only Cl emission inventory currently available for China. Diurnal, day-of-week, and vertical allocations of the emissions followed the methods introduced by Zhang et al. (2015). For natural emissions, the Model of Emissions of Gases and Aerosols from Nature version

2.1 (MEGAN) (Guenther et al., 2006) was used to calculate the biogenic emissions over the domain throughout the simulation period.

### 2.2.3 $O_3$ and $NO_2$ measurement data

Real-time measurements of ground-level $O_3$ and $NO_2$ were conducted routinely at ~1000 stations in the national air quality monitoring network that was established by the Ministry of Environmental Protection (MEP) of China since 2013. The MEP

has operated this monitoring network and made the data publically available at http://106.37.208.233:20035/ since 2013. The measurements were conducted at local environmental protection bureaus in each city following the same standards for instrument operation and quality control set by the China MEP (http://www.mep.gov.cn/). Hourly data from 908 surface stations were available during our simulation period and used in this study (Figure 1b). It should be noted that $NO_2$ measurements in MEP's network, as in regulatory networks of other countries, were made with the catalytic conversion of

$NO_2$ to NO. This method is known to overestimate $NO_2$, especially during the photochemically active daytime and in locations away from the sources of emissions (Xu et al., 2013).



# 3    Results and Discussion

## 3.1    Simulated HONO, $N_2O_5$, and $ClNO_2$ with the CBMZ_ReNOM over China

### 3.1.1 Spatial and vertical distributions of HONO

In Figures 2a and 2b, we show the spatial distributions of the surface $NO_2$ and HONO concentrations, respectively, simulated

by the CBMZ_ReNOM module. It can be observed that high levels of HONO were simulated over the NCP and YRD regions, with values of 800–1400 ppt and 1000–2000 ppt, respectively (Figure 2b). Over central China (mostly Jiangxi and Hubei provinces) and the PRD, HONO reached up to 1000 ppt during the simulated period. Similar spatial distributions of simulated HONO were also reported by Tang et al. (2015), who considered several heterogeneous sources, traffic emissions, and an unknown source of HONO, and simulated 0.5–2.5 ppb of HONO over the NCP, YRD, and PRD regions during two

summer seasons (Jul 2006 and Aug 2007). An interesting phenomenon is that high concentrations of HONO are simulated in Taiwan, Korea, and southern Japan. There may be two reasons for these results. Firstly, these regions had high $NO_x$ emissions and high $NO_2$ concentrations (up to 30 ppb during the simulation period, as illustrated in Figure 2a and Figure S1). Second, these regions are covered with large areas of vegetation that are mapped with high soil bacterial emissions (agriculture lands, shrubs, and woody lands), as measured by Oswald et al. (2013). Comparisons between the model

simulations of HONO and the limited observations in these regions will be discussed in section 3.2.

Figures 3a and 3b illustrate the vertical distributions of $NO_2$ and HONO across the NCP region and central China, respectively, during nighttime. The model predicted elevated HONO concentrations with values of 350–700 ppt over the NCP region and central China where high $NO_2$ concentrations are present. It can be seen in Figures 3b and 4b that the modeled HONO was mostly concentrated near the surface (0–200 m) within the PBL over the three most concerned regions

(NCP, YRD, and PRD) in China. This is due to two reasons. One is that $NO_2$, the main precursor of HONO, is highly concentrated within this layer (Figure 4a) and the heterogeneous formation at ground surface is the dominant source of HONO in the atmosphere as suggested in previous studies (Zhang et al., 2016; Li et al., 2010). The other reason is that the direct HONO emissions, including the soil bacterial emission and traffic emission, were also within the near-surface layer.

### 3.1.2 Spatial and vertical distributions of $N_2O_5$ and $ClNO_2$

Figures 2c and 2d show the simulated spatial distributions of $N_2O_5$ and $ClNO_2$ over East Asia. It can be seen that $N_2O_5$ was highly concentrated in the NCP and YRD regions, with levels of 20–60 ppt at the surface level and 30–100 ppt within the PBL (Figure 3c) over the polluted NCP region and central China during nighttime. Higher $N_2O_5$ levels were found in the residual layer (of 100–500 m a.g.l.) over the NCP and YRD regions (see Figure 4c). Low $N_2O_5$ concentrations were simulated over southern China, which is consistent with generally good air quality due to influence of monsoon weather in

the summer season (Wang et al., 2009). High levels of $ClNO_2$ in a range of 500–900 ppt were simulated in the NCP and YRD regions, where large amounts of chloride and $NO_x$ are emitted. The simulated concentrations of $ClNO_2$ in the NCP



were generally similar to the results reported by Sarwar et al. (2014), who used a hemispherical chemical transport model with a spatial resolution of 108 km; however our study gave more refined results at 27 km. Strong heterogeneous production of ClNO$_2$ was also simulated over central China (CC) and Sichuan Basin (SB) with intensive anthropogenic emissions of NO$_x$ and chloride. Up to 800 ppt and 600 ppt of ClNO$_2$ were simulated over the polluted regions (e.g. NCP, YRD, and CC) at the surface and spreading over the boundary layer (Figures 3d and 4d, respectively). The simulated ClNO$_2$ in the Hong Kong-PRD region was lower than those in the NCP, YRD, CC, and SB regions.

## 3.2  Model performance of HONO and N$_2$O$_5$/ClNO$_2$

We compared the simulations of HONO by the new module with field observations reported in Korea, Japan, Taiwan, Hong Kong, and mainland China (see Table 4). As listed, the measured HONO ranged from 0.59 to 2.8 ppb at multiple sites over China. Over the NCP, YRD, and PRD regions, the model simulations were generally in line with the observed HONO levels, especially those in summer seasons (Table 4). The model very well captured the measured HONO at Wangdu in the NCP region during a matching simulation period, reproducing 86% of the observations (0.81 vs. 0.94 ppb), and the simulations agreed well with the observations in coastal cities in northern Japan (Sapporo), southeast Japan (Tokushima), and Tokyo. Our simulations were consistent with these limited measurements in high spots (Figure 2b) in Taiwan, Japan, and Korea (Figure 2b). For instance, Tsai et al. (2014) measured HONO in multiple seasons during 2005-2007 using MOUDI in southern Taiwan (Kaohsiung) and found a HONO concentration of ~2.8 ppb at night and ~1.5 ppb during the day. Simulations of our model (3.3 ppb during nighttime and 1.9 ppb during daytime) were very close to the observations at this site. Kim et al. (2015) observed HONO concentrations up to 1 ppb in a forested area in Seoul, Korea, during the summer and Song et al. (2009) reported HONO concentrations with a maximum value of 8.61 ppb on the campus of the University of Seoul, Korea, between May and July, 2005. The model oversimulated the average HONO in Korea by 3–7 times, but it was able to capture the maximum observed values. The overestimates were probably due to uncertainties in the treatment of soil bacterial emissions over this region. Compared with the model performance reported in previous numerical studies, which reproduced 20–80% of the observed HONO (An et al., 2013; Li et al., 2010; Gonçalves et al., 2012; Elshorbany et al., 2012), the overall performance of HONO of the new module in our model was satisfactory.

Very limited observations of N$_2$O$_5$ and ClNO$_2$ have been made in Asia. In Table 4, we summarize the available measurements of N$_2$O$_5$ and ClNO$_2$ in Asia, along with the simulated values by our model. As shown, N$_2$O$_5$ with a mean value of 7–28 ppt was observed in Mt. Tai, an urban area (Shandong University in Jinan) and a semi-rural area (Wangdu) over the NCP region. In Japan, N$_2$O$_5$ measurements in Toyokawa were reported by Nakayama et al. (2008), with values of ~20 ppt. The observed ClNO$_2$ values were in the range of 30–160 ppt in Wangdu, Mt. Tai, and Jinan in the northern China. Overall, the simulated values by our new module were of the same order of magnitude as the observed values at the sites in the northern China and Japan. During the simulation period, the CBMZ_ReNOM module very well reproduced the average levels of N$_2$O$_5$ (observation of 27.9 vs. simulation of 23.9 ppt) and ClNO$_2$ (159.5 ppt vs. 265.6 ppt) observed at the Wangdu





site in northern China. Compared with the performance of previous model simulations, our results for $N_2O_5$ and $ClNO_2$ with the newly developed CBMZ_ReNOM module in China were satisfactory. For example, Lowe et al. (2015) oversimulated $N_2O_5$ by 15–32% due to the uncertainties in meteorological simulations and the treatment of the heterogeneous uptake of $N_2O_5$. Sarwar et al. (2014) reproduced the observed peak values of $ClNO_2$ by 12–2100% at multiple sites in the northern hemisphere, which was probably because of the relatively low model resolution (108 km). Li et al. (2016) simulated lower $N_2O_5$ levels within a factor of 3 and higher $ClNO_2$ within a factor of 4 compared to the observations at night in Hong Kong, which were partially due to the underestimated production of $N_2O_5$ and the overestimated uptake of $N_2O_5$.

### 3.3 Integrated effects on $O_3$ in the CBMZ_ReNOM

### 3.3.1 Enhancements in regional $RO_x$ and $O_3$ levels over polluted regions

Figures 5 and 6 show the spatial patterns at the surface (0-30 m) and the vertical distributions of daytime $RO_x$ and $O_3$ enhancement, respectively, over the three most concerned regions (NCP, PRD, and YRD), respectively. It is noticeable that the added $ClNO_2$ and HONO processes enhanced daytime $RO_x$ (OH+HO$_2$+RO$_2$) across the whole domain. Over the NCP and YRD regions, where the simulated $ClNO_2$ was elevated (Figure 5c), the enhancement in $RO_x$ due to the added chlorine chemistry was up to 1.8 ppt (12.5%) within the PBL (0–1000 m), and it increased with increasing altitude within the 0–500 m layer (see Figures 6a–6f). The releases of reactive Cl atoms and $NO_x$ through the daytime photolysis of $ClNO_2$, which serves as a reservoir of $NO_x$ during nighttime, led to more $O_3$ production, especially over the NCP and YRD regions. As shown in Figures 6g–6i, $ClNO_2$ increased the surface $O_3$ in the NCP and YRD regions by 2.4–3.3 ppb (or 5–6%). It also had a significant impact (3-6%) on the 200–1000 layer within the PBL, as also seen in Figure S3a. However, over the PRD region where $ClNO_2$ was less abundant during the simulated summer period (Figure 2d), the increases in $RO_x$ and $O_3$ were much smaller.

In comparison, HONO, which is mostly concentrated within the near-surface layer, significantly affected the total $RO_x$ and $O_3$ levels at the surface (Figure 6 and Figure S3a), with enhancements of 2.8–4.6 ppt (28–37%) and 2.9–6.2 ppb (6–13%), respectively, in the daytime averages of $RO_x$ and $O_3$ over the three most-developed regions. It had much smaller or even negative impacts above 300 m a.g.l. The simulated $O_3$ slightly decreased in some remote regions that were considered $NO_x$-limited regions (Figure 5f and Figure S3a) because the heterogeneous formation of HONO (the major source of HONO) would consume $NO_2$ and lead to less $O_3$ production in these $NO_x$-limited regions. As shown in Figures S1 and S2, the regional average of $NO_2$ decreased by 17% in the ReNOM case as compared with the base case over eastern China.

The combined effects of HONO and $ClNO_2$ on surface $O_3$ in the ReNOM case are illustrated in Figure 5c, which shows that $O_3$ was significantly enhanced by 8–10 ppb (10–15%) in the three developed regions, by 4–8 ppb in Hubei province in central China, and by 6–8 ppb in Sichuan province in southwestern China. On average, $ClNO_2$ and HONO increased the regional $O_3$ levels in the NCP, YRD, and PRD regions by 11.5%, 13.5%, and 13.3%, respectively, at the surface and by





7.8%, 7.1%, and 8.9% within the PBL, respectively (Figures 6j–6l). The simultaneous consideration of HONO and Cl-chemistry led to a larger $O_3$ enhancement compared to the summation of the effect of each chemistry (8.6% in the ReNOM case compared to 2.0% in the ReNOM_Cl case and 5.2% in the ReNOM_HONO case in daily $O_3$ enhancement over the whole domain). That is possibly because consideration of $N_2O_5/ClNO_2$ in the ReNOM case extended the cycling time of

$NO_x$, and the photolysis of $ClNO_2$ released additional $NO_2$ during the day, which enhanced the heterogeneous formation of HONO (HONO increased by ~17% over eastern China in the ReNOM case compared to the ReNOM_HONO case; see Figure S2), amplifying the effects of HONO on the formation of $O_3$.

### 3.3.2 Improvement of $O_3$ predictions over China

In this section, we compare the simulated surface $O_3$ concentrations with observations during the simulation period to

demonstrate the overall improvement of the revised WRF-Chem in simulating ground-level $O_3$. The results are shown in Table 5. There were 205, 141, and 67 monitoring stations available for the NCP, YRD, and PRD regions, respectively, and ~495 sites in other regions of China during the simulation period. It can be clearly seen that the default WRF-Chem model under-predicted the observed $O_3$ in the NCP and PRD regions by 9.9 ppb and 9.4 ppb, respectively, on average, whereas over-predicted the $O_3$ by 5.2 ppb at stations over the YRD region (also see Figure 7). The over-predictions over the YRD

region were possibly due to overestimating the anthropogenic $NO_x$ emission in the MEIC over this area. Annual $NO_x$ emission estimated by the MEIC in 2010 over the YRD region (including Jiangsu, Shanghai, and Zhejiang provinces) was much larger than the estimate by Fu et al. (2013) with a much higher resolution ($4 \times 4$ km) over the same region (3749 vs. 2777 kt in annual average). The uncertainties in the anthropogenic emissions (and other model inputs) would obviously affect the simulations of $O_3$ with WRF-Chem (Ahmadov et al., 2015). However, discussion on these uncertainties is beyond

the scope of this study. Compared with the surface observations of $NO_2$ at the MEP's network, our model over-predicted the averaged $NO_2$ over the YRD region (see Figure S1 in the supplementary information). Over all of China (~908 sites), the default WRF-Chem model simulated daily $O_3$ of 31.0 ppb in the base case, compared with the observed value of 35.3 ppb.

When considering Cl-chemistry alone, the model noticeably improved the $O_3$ prediction over China by reducing the mean bias by 1.5 ppb (4.4%) in the ReNOM_Cl case. The inclusion of HONO (in the ReNOM_HONO case) alone provided a

larger improvement in $O_3$ prediction, reducing the normalized mean bias (NMB) from -12.2% to -3.5%. By considering the two components in the new CBMZ_ReNOM module, the mean simulated $O_3$ concentrations at monitoring stations all over China increased from 31.0 ppb in the base case to 35.5 ppb in the ReNOM case, much closer to the averaged observations and with significant decreases in mean bias (decreased from -4.3 ppb to 0.1 ppb) and NMB (reduced from -12.2% to 0.4%) and with a modest improvement in correlation between the simulated and observed $O_3$ levels (see Table 5). These results

indicate a considerable improvement in the ability of the WRF-Chem model to simulate ground-level $O_3$ and possibly other secondary pollutants.





## 4   Summary and conclusions

In this study, we incorporated comprehensive processes of HONO and chlorine chemistry into a new chemical mechanism option, CBMZ_ReNOM, in the WRF-Chem model and applied the new model to simulating the spatial distribution of HONO, $ClNO_2$, and $N_2O_5$ and their impact on $O_3$ in China during the summer. Model simulations with the new module

indicated that HONO was concentrated over the Northern China Plain (NCP), the Yangtze River Delta (YRD), and the Pearl River Delta (PRD) regions, with levels of 800–1800 ppt at ground-level, whereas the simulated $N_2O_5$ and $ClNO_2$ were most abundant within the 0–600 m layer, with average concentrations of 100–160 ppt and 800–1200 ppt, respectively, over the NCP, YRD, central China, and Sichuan Basin. The combined processes of HONO and chlorine chemistry increased $RO_x$ mixing ratios by 36.3–44.7% at the surface and 4–37% within the PBL during the simulation period in summer and

enhanced the daytime $O_3$ levels over the NCP, YRD, and PRD regions by 11.5–13.5% (2.9–6.5 ppb) at the surface and up to 10.9% in upper levels within the PBL. HONO had a more obvious impact on daytime $O_3$ at the surface and near-surface layer, whereas $ClNO_2$ showed significant influence above ~300 m a.g.l. over the NCP and YRD regions. The revised WRF-Chem considerably improved $O_3$ prediction across China. Our results suggest the importance of HONO and $ClNO_2$ in the formation of $O_3$ in the lower troposphere over polluted regions in China, and underscore the need for considering these

reactive nitrogen species in chemical transport models to better predict ozone and other secondary pollutants.

*Acknowledgements*. The authors would like to thank the China Meteorological Administration for providing the meteorological observations and the China Ministry of Environmental Protection Ministry for the $O_3$ and $NO_2$ observations. This work was financially supported by the Hong Kong Research Grant Council (project C5022-14G and PolyU

153042/15E), the National Natural Science Foundation of China (41275123 and 91544213), and the Hong Kong Polytechnic University (1-ZE13). Both the data and source codes of the revised model used in this study are available from the authors upon request (cetwang@polyu.edu.hk).

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





**Table 1.** Mechanism of chlorine chemistry in the CBMZ_ReNOM in WRF-Chem.

| No. | Reaction | Reaction rate | Ref |
|---|---|---|---|
| 01 | $Cl_2 + hv \rightarrow Cl + Cl$ | $J_{Cl_2}$ | a |
| 02 | $HOCl + hv \rightarrow Cl + OH$ | $J_{HOCl}$ | a |
| 03 | $ClNO_2 + hv \rightarrow Cl + NO_2$ | $J_{ClNO_2}$ | a |
| 04 | $ClONO_2 + hv \rightarrow Cl + NO_3$ | $0.83 \times J_{ClONO_2}$ | a,b |
| 05 | $ClONO_2 + hv \rightarrow ClO + NO_2$ | $0.17 \times J_{ClONO_2}$ | a,b |
| 06 | $FMCl + hv \rightarrow Cl + CO + HO_2$ | $J_{FMCl}$ | a |
| 07 | $FMCl + OH \rightarrow Cl + CO + H2O$ | $5.0 \times 10^{-13}$ | a |
| 08 | $HCl + OH \rightarrow Cl + H_2O$ | $ARRH(1.7 \times 10^{-12}, -230.0)$ | a |
| 09 | $Cl_2 + OH \rightarrow HOCl + Cl$ | $ARRH(3.6 \times 10^{-12}, -1200.0)$ | a |
| 10 | $HOCl + OH \rightarrow ClO + H_2O$ | $5.0 \times 10^{-13}$ | a |
| 11 | $ClO + ClO \rightarrow 0.3\ Cl_2 + 1.4\ Cl + O_2$ | $1.63 \times 10^{-14}$ | a |
| 12 | $ClO + NO_2 \rightarrow ClONO_2$ | $7.0 \times 10^{-11}$ | a |
| 13 | $ClO + NO \rightarrow Cl + NO_2$ | $ARRH(6.4 \times 10^{-12}, 290.0)$ | a |
| 14 | $ClO + HO_2 \rightarrow HOCl + O_2$ | $ARRH(2.7 \times 10^{-12}, 220.0)$ | a |
| 15 | $ClO + OH \rightarrow HO_2 + Cl$ | $1.8 \times 10^{-11}$ | a |
| 16 | $ClO + OH \rightarrow HCl + O_2$ | $1.2 \times 10^{-12}$ | a |
| 17 | $Cl + O_3 \rightarrow ClO + O_2$ | $ARRH(2.3 \times 10^{-11}, -200.0)$ | a |
| 18 | $Cl + NO_2 \rightarrow ClNO_2$ | $TROE(1.8 \times 10^{-31}, 2.0, 1.0 \times 10^{-10}, 1.0)$ | a |
| 19 | $Cl + HO_2 \rightarrow HCl + O_2$ | $3.5 \times 10^{-11}$ | a |
| 20 | $Cl + HO_2 \rightarrow ClO + OH$ | $ARRH(7.5 \times 10^{-11}, -620.0)$ | a |
| 21 | $Cl + H_2O_2 \rightarrow HCl + HO_2$ | $ARRH(1.1 \times 10^{-11}, -980.0)$ | a |
| 22 | $Cl + NO_3 \rightarrow NO_2 + ClO$ | $2.4 \times 10^{-11}$ | a |
| 23 | $Cl + ClONO_2 \rightarrow Cl_2 + NO_3$ | $ARRH(6.2 \times 10^{-12}, 145.0)$ | a |
| 24 | $Cl + CH_4 \rightarrow HCl + CH_3O_2$ | $ARRH(6.6 \times 10^{-12}, -1240.0)$ | a |
| 25 | $Cl + C_2H_6 \rightarrow HCl + 0.991\ ALD2 + XO_2 + HO_2$ | $ARRH(8.3 \times 10^{-11}, -100.0)$ | a |
| 26 | $Cl + PAR \rightarrow HCl + XO_2 + 0.11\ HO_2$ $+ 0.06\ ALD2 + 0.11\ PAR + 0.76\ RO_2$ | $5.0 \times 10^{-11}$ | a |
| 27 | $Cl + ETH \rightarrow FMCl + 2\ XO_2 + HO_2 + HCHO$ | $1.07 \times 10^{-10}$ | a |
| 28 | $Cl + OLE \rightarrow FMCl + 0.33\ ALD2 + 2\ XO_2 + HO_2$ $+ PAR$ | $2.5 \times 10^{-10}$ | a |



| | | | |
|---|---|---|---|
| **29** | Cl + OLI → 0.3 HCl + 0.7 FMCl + 0.45 ALD2 + 0.3 OLE + 0.3 PAR + 1.7 XO2 + HO2 | $3.5 \times 10^{-10}$ | a |
| **30** | Cl + ISOP → 0.15 HCl + XO2 + HO2 + 0.85 FMCl+ISOPRD | $4.3 \times 10^{-10}$ | a |
| **31** | Cl + HCHO → HCl + HO2 + CO | ARRH($8.2 \times 10^{-11}$, -34.0) | a |
| **32** | Cl + ALD2 → HCl + C2O3 | $7.9 \times 10^{-11}$ | a |
| **33** | Cl + CH3OH → HCl + HO2 + HCHO | $5.5 \times 10^{-11}$ | a |
| **34** | Cl + ANOL → HCl + HO2 + ALD2 | ARRH($8.2 \times 10^{-11}$, 45.0) | a |
| **35** | Cl + TOL → HCl + 0.88 XO2 + 0.88 HO2 + 0.12 NAP | $6.1 \times 10^{-11}$ | c |
| **36** | Cl + XYL → HCl + 0.84 XO2 + 0.84 HO2 + 0.16 NAP | $1.2 \times 10^{-10}$ | d |

[a] The kinetic data are taken from the IUPAC database (http://iupac.pole-ether.fr/index.html); [b] The branching ratio is determined based on Tropospheric Ultraviolet Visible (TUV) Radiation model calculations; [c] Smith et al. (2002); [d] Wallington et al. (1988).

5  $\text{ARRH}(\alpha, \beta) = \alpha \times \text{EXP}(-\frac{\beta}{T})$ ; $\text{TROE}(k_0^{300}, n, k_\infty^{300}, m) = (\frac{k_0(T)[M]}{1+\frac{k_0(T)[M]}{k_\infty(T)}}) \times 0.6^{\left\{1+\left[\log_{10}(\frac{k_0(T)[M]}{k_\infty(T)})\right]^2\right\}^{-1}}$ , where $k_0(T) = k_0^{300} \times$

$(\frac{T}{300})^{-n}$ ; $k_\infty(T) = k_\infty^{300} \times (\frac{T}{300})^{-m}$; M is the number density; and T is the absolute temperature.



**Table 2.** WRF-Chem module configurations.

| Major Modules | Option | Reference |
|---|---|---|
| Microphysics scheme | Lin | *Lin et al. (1983)* |
| Cumulus scheme | Grell and Dévényi | *Grell and Dévényi (2002)* |
| Longwave radiation | RRTM | *Mlawer et al. (1997b)* |
| Shortwave radiation | Goddard shortwave | *Chou et al. (1998)* |
| Land-surface physics | Noah LSM | *Chen and Dudhia (2001)* |
| Urban surface scheme | UCM | *Kusaka et al. (2001)* |
| PBL scheme | MYJ | *Janjić (1994)* |
| Photolysis scheme | Fast-J | *Fast et al. (2006)* |
| Chemical mechanism | CBMZ/CBMZ_ReNOM | *Zaveri and Peters (1999)* |
| Aerosol module | MOSAIC | *Zaveri et al. (2008); Archer-Nicholls et al. (2014)* |



**Table 3.** Simulation cases of WRF-Chem model.

| Case | Chemical Mechanism | HONO Chemistry | ClNO$_2$ Chemistry |
|------|--------------------|----------------|--------------------|
| BASE | CBMZ | None | None |
| ReNOM_Cl | CBMZ_ReNOM | None | Yes |
| ReNOM_HONO | CBMZ_ReNOM | Yes | None |
| ReNOM | CBMZ_ReNOM | Yes | Yes |





**Table 4**. Comparisons between simulated and observed HONO (ppb), $N_2O_5$ (ppt), and $ClNO_2$ (ppt) over Asia from previous studies.

| Species | Location | Observation Period | Observation average | Simulation average | Reference |
|---|---|---|---|---|---|
| HONO | Wangdu, NCP[a] | Jun-Jul 2014 | 0.94 | 0.81 | Liu et al. (2016) |
| | Beijing (PKU), NCP[a] | Aug 2007 | 1.47 | 2.03 | Spataro et al. (2013) |
| | Beijing (Yufa), NCP[a] | Aug 2006 | 0.76 | 1.07 | Yang et al. (2014) |
| | Shanghai, YRD[a] | Oct-Jan 2004/05 | 1.10 | 1.15 | Hao et al. (2006) |
| | Hong Kong (TC), PRD[a] | Aug 2011 | 0.92 | 0.78 | Zhang et al. (2016) |
| | Xinken, PRD[a] | Oct-Nov 2004 | 1.20 | 0.18 | Su et al. (2008) |
| | Guangzhou, PRD[a] | Jul 2006 | 2.80 | 1.49 | Qin et al. (2009) |
| | Backgarden, PRD[a] | Jul 2006 | 0.59 | 0.83 | Li et al. (2012) |
| | Taehwa, Korea | Jun 2012 | 0.60 | 2.26 | Kim et al. (2015) |
| | Seoul, Korea | May-Jul 2005 | 0.36 (max 8.6) | 2.75 (max 8.7) | Song et al. (2009) |
| | Tokyo, Japan | Jan-Feb 2004 | 0.43 | 0.45 | Kanaya et al. (2007) |
| | Tokushima, Japan | Aug 2011 | 0.63 | 0.53 | Takeuchi et al. (2013) |
| | Tokushima, Japan | Feb 2011 | 0.56 | 0.53 | Takeuchi et al. (2013) |
| | Sapporo, Japan | Oct 2002 | 0.98 | 0.47 | Noguchi et al. (2010) |
| | Kaohsiung, Taiwan | 2005-2007 (multiple months) | 2.13 | 2.58 | Tsai et al. (2014) |
| | Taichung, Taiwan | Jan-Dec 2002 | 1.53 | 2.23 | Lin et al. (2006) |
| $N_2O_5$ | Wangdu, NCP[a] | Jun-Jul 2014 | 28.0 | 23.9 | Tham et al. (2016) |
| | Mt. Tai, NCP[a] | Jul-Aug 2014 | 7.00 | 18.3 | *unpublished data* |
| | Jinan, NCP[a] | Aug-Sep 2014 | 17.0 | 13.1 | *unpublished data* |
| | Hong Kong (TMS), PRD[a] | Nov-Dec 2013 | 277.8 (nighttime) | 6.27 (nighttime) | Wang et al. (2016) |
| | Toyokawa, Japan | Feb 2006 | 20.0 (max) | 14.1 (max) | Nakayama et al. (2008) |
| $ClNO_2$ | Wangdu, NCP[a] | Jun-Jul 2014 | 159.5 | 265.6 | Tham et al. (2016) |



| | | | | |
|---|---|---|---|---|
| Mt. Tai, NCP[a] | Jul-Aug 2014 | 30.4 | 117.4 | *unpublished data* |
| Jinan, NCP[a] | Aug-Sep 2014 | 94.0 | 254.0 | *unpublished data* |
| Hong Kong (TMS), PRD[a] | Nov-Dec 2013 | 74.6 (nighttime) | 8.69 (nighttime) | Wang et al. (2016) |

[a] NCP: Northern China Plain; YRD: Yangtze River Delta; PRD: Pearl River Delta; PKU: Peking University; TC: Tung Chung; TMS: Tai Mo Shan.



**Table 5.** Statistics of model performance in the base and ReNOM cases for hourly O$_3$ measurements (ppb) at ~908 MEP air quality monitoring stations during the simulation period (27 Jun-7 Jul 2014).

| Region | Case | No. OBS[b] | OBS[b] | MOD[b] | COR[b] | MB[b] | RMSE[b] | NMB[b] | NME[b] |
|---|---|---|---|---|---|---|---|---|---|
| NCP[a] | BASE | 49789 | 47.2 | 37.3 | 0.60 | -9.9 | 26.8 | -20.9% | 44.0% |
| | ReNOM_Cl | | | 40.0 | 0.61 | -7.2 | 26.7 | -15.3% | 43.7% |
| | ReNOM_HONO | | | 41.2 | 0.61 | -6.0 | 26.4 | -12.7% | 43.3% |
| | ReNOM | | | 43.5 | 0.61 | -3.6 | 26.5 | -7.7% | 43.5% |
| YRD[a] | BASE | 34857 | 31.5 | 36.8 | 0.56 | 5.2 | 29.3 | 16.6% | 68.2% |
| | ReNOM_Cl | | | 39.4 | 0.56 | 7.9 | 31.9 | 25.1% | 72.7% |
| | ReNOM_HONO | | | 43.3 | 0.54 | 11.8 | 34.3 | 37.3% | 77.5% |
| | ReNOM | | | 45.3 | 0.54 | 13.8 | 36.3 | 43.8% | 81.7% |
| PRD[a] | BASE | 15627 | 25.0 | 15.6 | 0.53 | -9.4 | 28.0 | -37.6% | 75.8% |
| | ReNOM_Cl | | | 15.6 | 0.53 | -9.4 | 27.9 | -37.6% | 75.8% |
| | ReNOM_HONO | | | 18.5 | 0.54 | -6.4 | 26.3 | -25.8% | 73.7% |
| | ReNOM | | | 18.4 | 0.54 | -6.5 | 26.3 | -26.1% | 73.9% |
| China | BASE | 214596 | 35.3 | 31.0 | 0.51 | -4.3 | 27.3 | -12.2% | 57.5% |
| | ReNOM_Cl | | | 32.6 | 0.51 | -2.8 | 28.0 | -7.8% | 58.5% |
| | ReNOM_HONO | | | 34.1 | 0.52 | -1.2 | 26.7 | -3.5% | 56.6% |
| | ReNOM | | | 35.5 | 0.52 | 0.1 | 26.1 | 0.4% | 55.8% |

[a] NCP: Northern China Plain; YRD: Yangtze River Delta; PRD: Pearl River Delta. [b] No. Obs: number of available observations used in evaluation; OBS: average observed value; MOD: average modeled value; COR: correlation; MB: mean bias; RMSE: root mean square error; NMB: normalized mean bias; NME: normalized mean error.



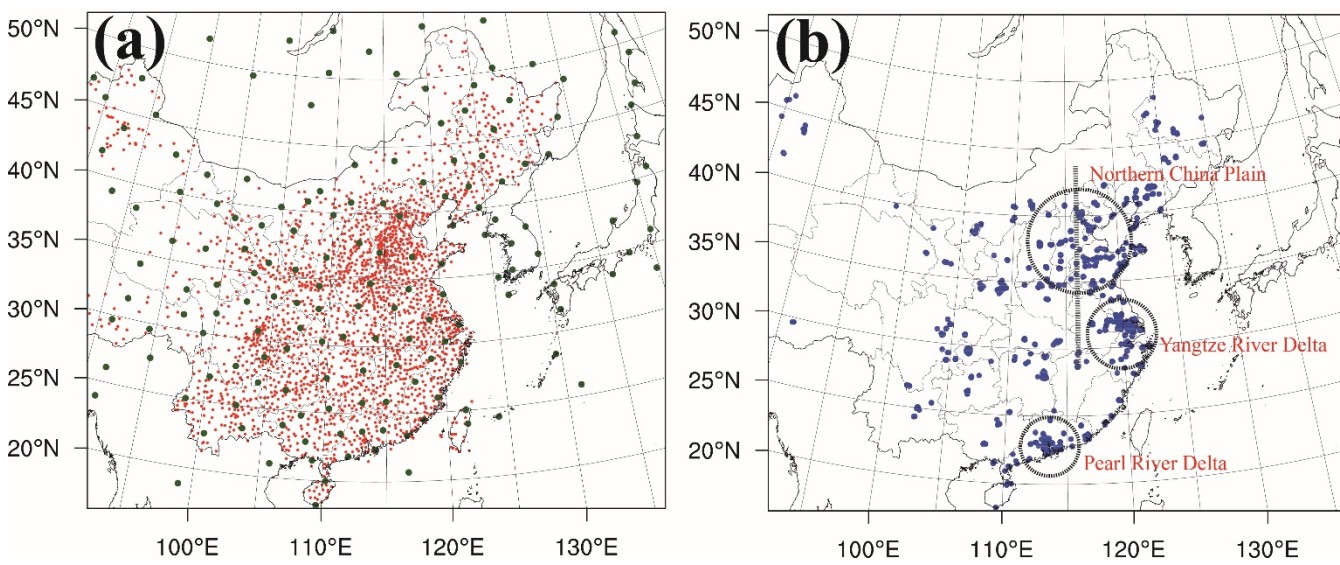

**Figure 1.** WRF-Chem domain used in this study. (a) Red dots denote the surface weather stations used in FDDA (tri-hourly); green dots are the sounding sites (12-hourly). (b) Blue dots denote the available surface air-quality monitoring stations operated by China MEP in 2014, and the dashed line represents the vertical domain that intercepts the most polluted Northern China Plain.







**Figure 2.** Spatial distributions of averaged (a) NO$_2$ (ppb), (b) HONO, (c) nighttime N$_2$O$_5$ (18:00–06:00 Local Time Coordinate (LTC)), and (d) nighttime ClNO$_2$ (ppt) in the ReNOM case at the surface (~30 m) during the simulation period in summer 2014.







**Figure 3.** Vertical distributions of (a) NO$_2$ (ppb), (b) HONO, (c) N$_2$O$_5$, and (d) ClNO$_2$ (ppt) during nighttime (18:00–06:00 LTC) in the domain intercepting northern China and central China. Vectors present the average v–w wind components (m s$^{-1}$), the dash lines the temperature (°C), and the black line the simulated planetary boundary layer height during the nighttime.





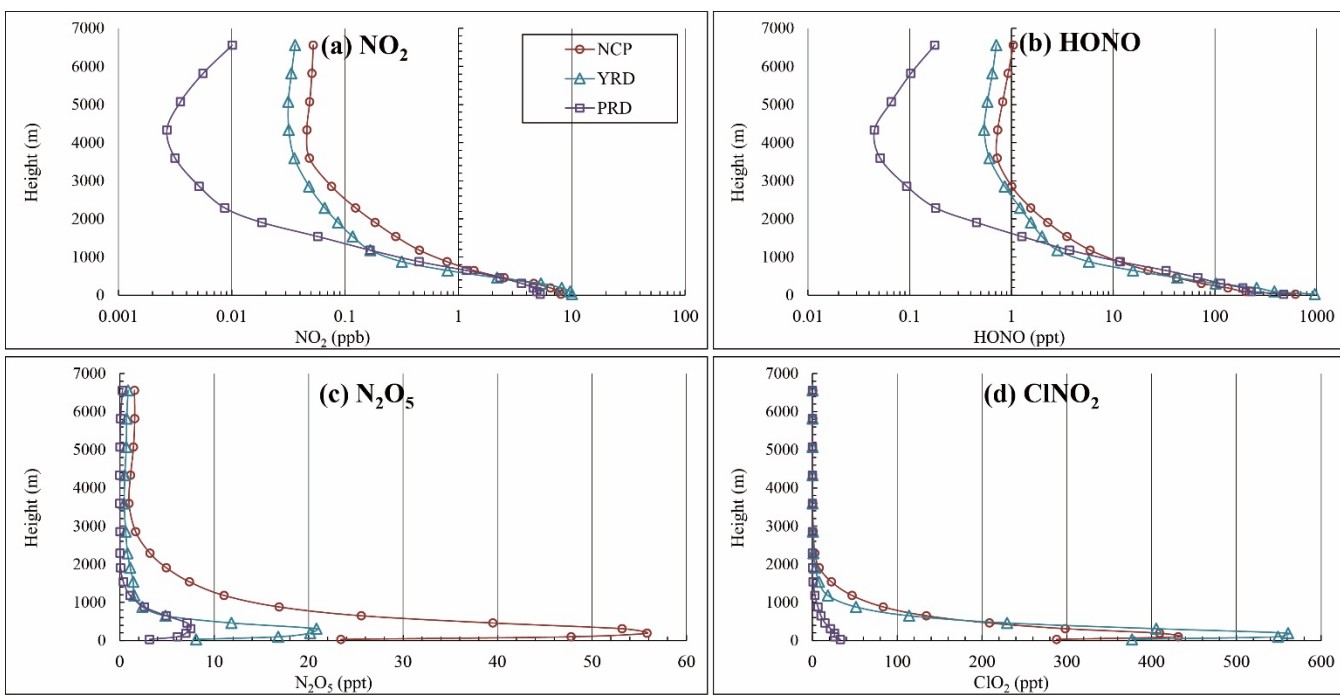

**Figure 4.** Vertical distributions of WRF-Chem simulated regional averages of (a) $NO_2$ (ppb), (b) HONO, (c) nighttime $N_2O_5$, and (d) nighttime $ClNO_2$ (ppt) over the NCP, YRD, and PRD regions.



5   **Figure 5.** Daytime (06:00–18:00 LTC) $RO_x$ enhancements (ppt) in (a) ReNOM_Cl case, (b) ReNOM_HONO case, and (c)

ReNOM case; $O_3$ enhancements (ppb) in (d) ReNOM_Cl case, (e) ReNOM_HONO case, and (f) ReNOM case.





5   **Figure 6.** Daytime (06:00–18:00 LTC) vertical $RO_x$ enhancements (ppt) over the (a) NCP, (b) YRD, and (c) PRD regions; $RO_x$ percentage enhancements (%) over the (d) NCP, (e) YRD, and (f) PRD regions; $O_3$ enhancements (ppb) over the (g) NCP, (h) YRD, and (i) PRD regions; $O_3$ percentage enhancements (%) over the (j) NCP, (k) YRD, and (l) PRD regions.



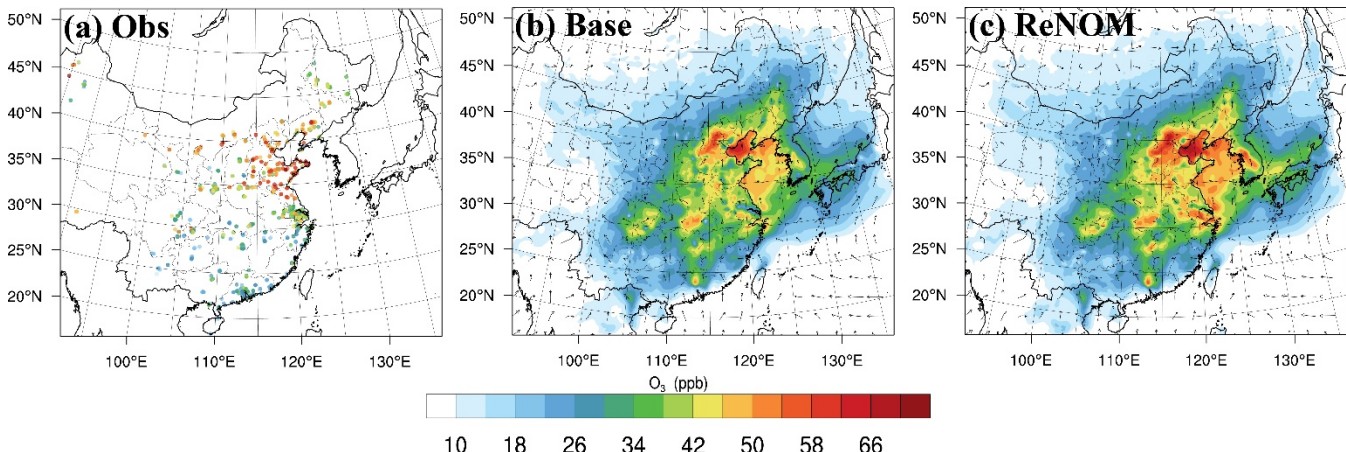

**Figure 7.** (a) Observations of O₃ at China MEP stations; spatial distributions of modeled O₃ concentrations in (b) base case and (c) ReNOM case (ppb).