# Peer review of "Combined Impacts of Nitrous Acid and Nitryl Chloride on Lower Tropospheric Ozone: New Module Development in WRF-Chem and Application to China"

_Atmospheric Chemistry and Physics, 2017_

## Referee Comment (RC1) · Anonymous Referee #1 · 23 May 2017

General comments The authors previously expanded the gas-phase CBMZ mechanism in WRF-Chem model to include comprehensive sources of HONO. Here, they update the CMBZ mechanism by incorporating HONO and chlorine chemistry including the heterogeneous ClNO2 formation. They perform four different model simulations over China for a 12-day period using a 27-km horizontal grid resolution, describe impact of the chemistry, compare model results with observed data and suggest that the additional chemistry increases HONO, ClNO2, and ozone and improves model performance. Overall, the article is written clearly and merits publication. However, several issues need to be addressed before publication.

Specific comments

Introduction, line 12-13 While chlorine atoms react with hydrocarbons, the reactions of NO2 with hydrocarbons are generally negligible. Clarifications are needed.

Heterogeneous formations, line 16-18 Foley et al. (2010) article does not provide any reaction rates for R5–R7. Correct reference is needed. It will be convenient to readers to include the reaction rates for R5–R7 in this article.

Direction emissions, line 20-23 HONO emissions prescribed as 1.6% of NO2 emissions appear to be too high. Generally, HONO emissions are prescribed as a function of NOx emissions. A reference is needed.

Model configurations, line 17-19 The Model for Ozone and Related Chemical Tracers likely did not contain any ClNO2 and/or additional HONO sources. The authors may include a sentence to clarify the issue.

Model configurations, line 24-25 The spin-up time of 24 h appears to be too small.

Emission data, line 3-6 SO2, NOx, CO, CO2, NH3, PM2.5, PM10, BC, OC are not defined anywhere.

Spatial and vertical distributions of N2O5 and ClNO2, line 30-31 A plot of chloride distribution will be helpful to readers.

Model performance of HONO and N2O5/ClNO2 NOx emissions affect HONO, ClNO2, as well as O3 production. Thus, NOx emissions are critical for this study. Authors present a qualitative comparison of observed and measured NO2 in Figure S1. Is it possible to calculate model performance for NO2 and present a table similar to Table 5?

Enhancements in regional ROX and O3 levels over polluted regions, line 11-12 ROx is defined in line 12 but used in line 10 prior to defining. It is good to define it at the time of first introduction.

Summary and conclusions, line 1-15 HONO production is related to the prescribed NOx emissions while ClNO2 production is related to the prescribed NOx and chloride emissions. A very brief discussion on the uncertainty of NOx and chloride emissions is needed.

Summary and conclusions, line 1-8 Simulations were conducted for a 12-day period, not for the entire summer. Thus, it is I suggest adding the first sentence as follows (or something similar it):

In this study, we incorporated comprehensive processes of HONO and chlorine chemistry into a new chemical mechanism option, CBMZ_ReNOM, in the WRF-Chem model and applied the new model to simulating the spatial distribution of HONO, ClNO2, and N2O5 and their impact on O3 in China during the 12-day simulation period in summer.

NCP, YRD, PRD have already been defined earlier; no need to redefine them.

Summary and conclusions, line 12-13 Model performance improved at NCP, PRD, and China but deteriorated at YRD (Table 5). Thus, some caveat is needed. Perhaps, the authors may revise the sentence as follows (or something similar): With current emissions estimates, the revised WRF-Chem generally improved O3 prediction across China.

Table 1 It appears that some of the references are not correct. For example, reaction 18 (Cl + NO2 = ClNO2 is not included in IUPAC). Please check all references and update as appropriate.

It appears that rate constant for reaction 12 (ClO + NO2 = ClONO2) is not taken from IUPAC. IUPAC recommends a pressure dependent rate constant.

All symbols need to be defined.

Figure S2. It will be helpful to readers to define eastern China. Perhaps, the authors can mark "eastern China" in Figure S1 or other figures.

---

## Referee Comment (RC2) · Anonymous Referee #2 · 26 May 2017

Zhang et al. describe an updated WRF-Chem model with a new chemical mechanism (CBMZ_ReNOM) developed to improve predictions of photochemical O3 production in Eastern China, incorporating revised source chemistry of HONO and photolyzable chlorine species.

The paper is written well and is publishable. Ultimately, I wasn't sure how much an improvement this work actually represents. One might hope that a more explicit representation of chemistry within a model does improve its accuracy. Perhaps this part of the manuscript could be strengthened, for instance, through a more reasonable

comparison of predicted with observed concentrations: instead of comparing averages (Tables 4 and 5), why not compare selected time series of measurements and model predictions, in particular for transient species such as HONO and ClNO2 whose concentrations are highly variable.

Specific comments

pg 3 line 17 "To the best of our knowledge, no global or regional models, however, have simultaneously considered the sources/processes of HONO and ClNO2 and evaluated their regional impacts on the formation of O3 pollution in the boundary layer of the atmosphere." I am not sure the statement as written is true. Many studies have simultaneously considered HONO and ClNO2 as radical sources and showed how these species affect O3. For instance, Sarwar et al 2004 [GRL, 2014] studied O3 formation using CMAQV5.02 which contains the RACM2 mechanism [Goliff et al., AE 2013] and HONO chemistry. Also, Ahmadov et al. [ACP, 2015] and Edwards et al. [Nature, 2014] used models and HONO/ClNO2 data to investigate high wintertime ozone pollution events in an oil- and natural gas-producing region of the western US. There have also been numerous papers using 0D box models examining this chemistry.

pg 4 line 9 "reproduced the observed HONO by 85% on average" Not sure what this means.

pg 5 line 17. What is FMCl? A fluorine-metal compound?

pg 7 line 7. "has been proved" A model cannot be proven, at least not in a mathematical sense. Why not simply say that this model has made reasonable predictions of PM2.5 and O3?

pg 7, section "2.2.3 O3 and NO2 measurement data" Please state how accurate these measurements are.

pg 7, last line. The Mo converter also "detects" NO3, 2N2O5, HONO, ClNO2, PAN, and HNO3 to some degree as if it were NO2. The model should give some indication

as what fraction of NOy is in the form of NOz (as a function of time of day); consider a sensitivity run in which the NO2 reported by the routine measurements is scaled down by this factor.

pg 8 " 3.1.2 Spatial and vertical distributions of N2O5 and ClNO2 ". These predicted concentrations are considerably lower than recent measurements in the HK area (see sections further down).

pg 9, line 11 "The model very well captured the measured HONO at Wangdu in the NCP region during a matching simulation period, reproducing 86% of the observations (0.81 vs. 0.94 ppb)" I don't understand this sentence. How does a model reproduce 86% of observations? Consider instead a scatter plot of model concentrations vs observations.

pg 9, line 25 onward. Brown, S. S., et al. (2016), Nighttime chemistry at a high altitude site above Hong Kong, J. Geophys. Res.-Atmos., 121(5), 2457-2475, doi: 10.1002/2015jd024566 observed much higher concentrations than the model predicts. Please discuss.

pg 18, Table 1. Please state the units of the reaction rates.

rxn 7. Water should have a subscript.

rxn 26. What are PAR and X? Also, define the other terms, such as OLI, ALD2 etc.
* * *

---

## Author Comment (AC1) · 25 Jun 2017

**General comments:**

The authors previously expanded the gas-phase CBMZ mechanism in WRF-Chem model to include comprehensive sources of HONO. Here, they update the CMBZ mechanism by incorporating HONO and chlorine chemistry including the heterogeneous ClNO2 formation. They perform four different model simulations over China for a 12-day period using a 27-km horizontal grid resolution, describe impact of the chemistry, compare model results with observed data and suggest that the additional chemistry increases HONO, ClNO2, and ozone and improves model performance. Overall, the article is written clearly and merits publication. However, several issues need to be addressed before publication.

**Response:** Thanks for the encouragement. We have revised our manuscript according to the helpful comments.

(Our replies to the comments are highlighted in blue here, and all changes in the revised manuscript are highlighted in red)

**Specific comments**

1. Introduction, line 12-13 While chlorine atoms react with hydrocarbons, the reactions of NO2 with hydrocarbons are generally negligible. Clarifications are needed.

**Response:** The sentence has been revised into 'ClNO$_2$ is photolyzed to recycle NO$_2$ and release reactive chlorine atoms (R4), the latter of which further react with hydrocarbons to produce additional peroxy radicals'

**2.** Heterogeneous formations, line 16-18 Foley et al. (2010) article does not provide any reaction rates for R5–R7. Correct reference is needed. It will be convenient to readers to include the reaction rates for R5–R7 in this article.

**Response:** As suggested, we have made corrections on the reference and added the reactions rates.

**3.** Direction emissions, line 20-23 HONO emissions prescribed as 1.6% of NO2 emissions appear to be too high. Generally, HONO emissions are prescribed as a function of NOx emissions. A reference is needed.

**Response:** We checked that we applied an emission ratio of 0.8% (HONO to NOx). Corrections have been made and a reference has been added.

**4.** Model configurations, line 17-19 The Model for Ozone and Related Chemical Tracers likely did not contain any ClNO2 and/or additional HONO sources. The authors may include a sentence to clarify the issue.

**Response:** As suggested, a sentence has been added to clarify this: 'Note that the MOZART model does not treat chlorine chemistry nor consider any HONO sources.'

**5.** Model configurations, line 24-25 The spin-up time of 24 h appears to be too small.

**Response:** Since we used global model simulations as initial and boundary conditions and the lifetimes of those reactive nitrogen compounds are relatively short, we think that a spin-up time of 24 h is acceptable.

**6.** Emission data, line 3-6 SO2, NOx, CO, CO2, NH3, PM2.5, PM10, BC, OC are not defined anywhere.

**Response:** Definitions of these species have been added as suggested.

**7.** Spatial and vertical distributions of N2O5 and ClNO2, line 30-31 A plot of chloride distribution will be helpful to readers.

**Response:** Very good suggestion. Plots of spatial and vertical distributions of chloride have been added in the supplementary materials (Figure S2).

**8.** Model performance of HONO and N2O5/ClNO2 NOx emissions affect HONO, ClNO2, as well as O3 production. Thus, NOx emissions are critical for this study. Authors present a qualitative comparison of observed and measured NO2 in Figure S1. Is it possible to calculate model performance for NO2 and present a table similar to Table 5?

**Response:** Here we only presented a qualitative for $NO_2$ because what the MEP monitoring network measured were $NO_2^*$ instead of real $NO_2$. The $NO_2$ measurements in the MEP's network, as in regulatory networks of other countries, were made with the catalytic conversion of $NO_2$ to NO. In addition to $NO_2$, the MoO converter would also PAN, $HNO_3$, $HO_2NO_2$, HONO, $ClNO_2$, etc. into NO, and these interferences would let the instrument "detect" higher $NO_2$. So the catalytic method was suggested to overestimate $NO_2$ by 6%-280%, especially during the photochemically active daytime and in locations away from the sources of emissions.

In order to give a more reliable quantitative comparison, we used the model simulated NOz to try to scale down the measured NO2* at each hour by using a factor: $NO_{2\ obs} = NO_{2\ obs}^* \times$

$$\frac{NO_{2\ mod}}{NO_{2\ mod}+NO_{z\ mod}-Nitrate_{mod}}.$$ (gas-phase $HNO_3$ is assumed to be converted into NO by 80% due to its loss on inlet (Xu et al., 2013); other gas-phase NOz species are assumed to be converted by 100% in the Mo converter). **Table S1** in the revised supplementary materials lists the comparisons between the simulated and measured (both original and adjusted) $NO_2$. The simulations agreed well with the adjusted $NO_2$ measurements over China, except for the Yangtze River Delta region. The revised **Figure S1** in the supplementary materials also shows the original and scaled measurements of $NO_2$.

**Table S1.** Statistics of model performance in the base and ReNOM cases for hourly $NO_2$ measurements (ppb) at the MEP air quality monitoring stations during the simulation period (27 Jun-7 Jul 2014).

| Region | Case | No. OBS | OBS[a] | MOD | COR | MB | RMSE | NMB | NME |
|---|---|---|---|---|---|---|---|---|---|
| NCP | BASE | 48362 | 19.4 | 18.4 | 0.35 | -1.0 | 16.3 | -5.0% | 58.6% |
| | ReNOM | | | 14.4 | 0.35 | -5.0 | 15.8 | -25.8% | 56.3% |
| YRD | BASE | 35421 | 17.7 | 27.0 | 0.21 | 9.3 | 23.5 | 52.3% | 96.9% |
| | ReNOM | | | 19.2 | 0.17 | 1.4 | 17.9 | 8.1% | 75.8% |
| PRD | BASE | 15651 | 12.1 | 8.3 | 0.38 | -3.9 | 11.9 | -31.8% | 71.5% |
| | ReNOM | | | 7.2 | 0.35 | -5.0 | 11.5 | -40.8% | 69.7% |
| China | BASE | 213308 | 15.6 | 14.2 | 0.32 | -1.4 | 16.3 | -9.2% | 73.1% |
| | ReNOM | | | 10.9 | 0.32 | -4.8 | 14.6 | -30.5% | 67.4% |

| Region | Case | No. OBS | OBS_scaled[a] | MOD | COR | MB | RMSE | NMB | NME |
|---|---|---|---|---|---|---|---|---|---|
| NCP | BASE | 48362 | 15.6 | 18.4 | 0.48 | 2.8 | 14.4 | 17.9% | 62.2% |
| | ReNOM | | 13.5 | 14.4 | 0.51 | 0.9 | 11.7 | 6.9% | 57.6% |
| YRD | BASE | 35421 | 14.4 | 27.0 | 0.37 | 12.6 | 23.3 | 87.3% | 113.9% |
| | ReNOM | | 11.7 | 19.2 | 0.38 | 7.5 | 16.6 | 64.4% | 92.7% |
| PRD | BASE | 15651 | 10.0 | 8.3 | 0.37 | -1.7 | 11.0 | -17.1% | 77.0% |
| | ReNOM | | 8.1 | 7.2 | 0.42 | -0.9 | 9.1 | -11.5% | 75.3% |
| China | BASE | 213308 | 12.3 | 14.2 | 0.43 | 1.9 | 14.7 | 15.2% | 77.6% |
| | ReNOM | | 10.5 | 10.9 | 0.45 | 0.4 | 11.3 | 3.7% | 70.7% |

[a] OBS: original observations of $NO_2$; OBS_scaled: scaled observations of $NO_2$ based on model simulated reactive nitrogen species by using the equation of $NO_{2\ obs} = NO_{2\ obs}^{*} \times$ $\frac{NO_{2\ mod}}{NO_{2\ mod}+NO_{z\ mod}-Nitrate_{mod}}$, where $NO_2{*}_{obs}$ is the original measurement of $NO_2$, $NO_{2\ mod}$ is the model simulation of $NO_2$, $NO_{Z\ mod}$ is the sum of simulations of HONO, $2\times N_2O_5$, $ClNO_2$, $ClONO_2$, $NO_3$, $HNO_3$, $HNO_4$, PAN, and Nitrate, $Nitrate_{mod}$ is the simulated nitrate; gas-phase $HNO_3$ is assumed to be converted into NO by 80% in the Mo converter due to its possible loss on inlet; other gas-phase $NO_Z$ species are assumed to be converted by 100%.

**9.** Enhancements in regional ROX and O3 levels over polluted regions, line 11-12 ROx is defined in line 12 but used in line 10 prior to defining. It is good to define it at the time of first introduction.

**Response:** Thanks. The definition of ROx has been added in section 3.3.1 when it was introduced for the first time.

**10.** Summary and conclusions, line 1-15 HONO production is related to the prescribed NOx emissions while ClNO2 production is related to the prescribed NOx and chloride emissions. A very brief discussion on the uncertainty of NOx and chloride emissions is needed.

**Response:** We have added the following brief discussion on the uncertainties of NOx and chlorine emissions as suggested in section 3.2: "Since the emissions of NOx, the main precursor of HONO and $N_2O_5$, are subject to uncertainties in terms of intensity and spatial distribution (e.g. the possible overestimates over the YRD as we discussed) and the chlorine emission provided by the RCEI that we applied in the present study is with large uncertainties due to its relatively low resolution and its temporal coverage being 1990, our model results of HONO and $ClNO_2$ (and their impacts) are certainly with uncertainties."

**11.** Summary and conclusions, line 1-8 Simulations were conducted for a 12-day period, not for the entire summer. Thus, it is I suggest adding the first sentence as follows (or something similar it): In this study, we incorporated comprehensive processes of HONO and chlorine chemistry into a new chemical mechanism option, CBMZ_ReNOM, in the WRF-Chem model and applied the new model to simulating the spatial distribution of HONO, ClNO2, and N2O5 and their impact on O3 in China during the 12-day simulation period in summer.

**Response:** Thank you very much for this suggestion. The sentence has been modified as suggested.

**12.** NCP, YRD, PRD have already been defined earlier; no need to redefine them.

**Response:** The definitions were deleted here.

**13.** Summary and conclusions, line 12-13 Model performance improved at NCP, PRD, and China but deteriorated at YRD (Table 5). Thus, some caveat is needed. Perhaps, the authors may revise the sentence as follows (or something similar): With current emissions estimates, the revised WRF-Chem generally improved O3 prediction across China.

**Response:** Very good suggestion. We have revised this conclusion sentence as suggested.

**14.** Table 1 It appears that some of the references are not correct. For example, reaction 18 (Cl + NO2 = ClNO2 is not included in IUPAC). Please check all references and update as appropriate.

**Response:** Thanks a lot for pointing this out. We have carefully checked all the references we used and made the corrections accordingly. Please refer to the revised Table 1 in the manuscript. The reaction rate for R18 was taken from Tanaka, et al. 2003 (Development of a chlorine mechanism for use in the carbon bond IV chemistry model, J. Geophys. Res.-Atmos., 108, 4145, 10.1029/2002JD002432, 2003).

**15.** It appears that rate constant for reaction 12 (ClO + NO2 = ClONO2) is not taken from IUPAC. IUPAC recommends a pressure dependent rate constant.

**Response:** It is true that the rate constant for this reaction in IUPAC database is pressure dependent. But we considered that those low-pressure rate coefficients (applicable for pressure ranging from 1.3 to 7 mbar) are not suitable for calculating this reaction in PBL, and thus we applied the preferred high-pressure rate value recommended in IUPAC, which is $7 \times 10^{-11}$ and is independent of temperature over the range 250-350 K (http://iupac.pole-ether.fr/htdocs/datasheets/pdf/iClOx32_ClO_NO2_M.pdf). This constant rate has been applied in our previous MCM model development study for chlorine chemistry (Xue, L. K. et al., Development of a chlorine chemistry module for the Master Chemical Mechanism, Geosci. Model Dev., 8, 3151-3162, 10.5194/gmd-8-3151-2015, 2015).

**16.** All symbols need to be defined.

**Response:** We have added the definitions of all symbols.

**17.** Figure S2. It will be helpful to readers to define eastern China. Perhaps, the authors can mark "eastern China" in Figure S1 or other figures.

**Response:** Thanks a lot for the suggestion. We now define the eastern China area in Figure S1.

---

## Author Comment (AC2) · 25 Jun 2017

Zhang et al. describe an updated WRF-Chem model with a new chemical mechanism (CBMZ_ReNOM) developed to improve predictions of photochemical O3 production in Eastern China, incorporating revised source chemistry of HONO and photolyzable chlorine species.

The paper is written well and is publishable. Ultimately, I wasn't sure how much an improvement this work actually represents. One might hope that a more explicit representation of chemistry within a model does improve its accuracy. Perhaps this part of the manuscript could be strengthened, for instance, through a more reasonable comparison of predicted with observed concentrations: instead of comparing averages (Tables 4 and 5), why not compare selected time series of measurements and model predictions, in particular for transient species such as HONO and ClNO2 whose concentrations are highly variable.

**Response:** We think that we have made a contribution to the development of this widely-used regional chemical transport model, WRF-Chem, which allows the model to be able to simultaneously consider the HONO and Chlorine chemistry. Besides, we evaluate the combined effects of HONO and chlorine chemistry on the lower tropospheric ozone in China at a regional scale which has been rarely reported before.

In addition to comparing the averages of simulations and observations, we did make a detailed comparison of modeled results and measurements during the CareBeijing 2014 campaign at Wangdu in the northern China. But we did not include this part into the manuscript to make this manuscript more concise and readable and also because that we had made detailed comparisons between modeled and measured HONO and $ClNO_2$ separately in our previous studies. We think the suggestion from Review 2 makes a good point. Therefore, we put these detailed comparisons during the CareBeijing 2014 campaign into the Supplement Information (details of the campaign can be found in Tham et al., 2016; Tan et al., 2017, and references therein). Please see the Table S2 listing the statistics of model performances for major pollutants and Figure S3 (in the revised supplementary materials) showing the time series of modeled and measured results.

**Table S2.** Observed and simulated major pollutants obtained from the CareBeijing 2014 campaign at Wangdu during the simulated period.

| Species | OBS | BASE | ReNOM_Cl | ReNOM_HONO | ReNOM |
|---|---|---|---|---|---|
| CO (ppb) | 541.0 | 577.4 | 578.4 | 572.3 | 574.2 |
| $SO_2$ (ppb) | 7.7 | 8.1 | 8.0 | 7.9 | 7.9 |
| $NO_2$ (ppb) | 12.8 | 13.5 | 12.4 | 11.4 | 10.7 |
| $O_3$ (ppb) | 55.6 | 51.5 | 54.6 | 55.5 | 56.5 |
| $PM_{2.5}$ ($\mu g/m^3$) | 84.9 | 90.5 | 101.6 | 96.8 | 106.6 |
| HONO (ppt) | 941.2 | 38.5 | 37.2 | 769.4 | 805.3 |
| $N_2O_5$ (ppt) | 28.0 | / | 28.0 | / | 23.9 |

| ClNO$_2$ (ppt) | 159.5 | / | 279.3 | / | 265.6 |
| --- | --- | --- | --- | --- | --- |

[Figure]

**Figure S3.** Observed and simulated (a) NO$_2$, (b) HONO, (c) N$_2$O$_5$ and (b) ClNO$_2$ at the Wangdu site during the simulation period (27 Jun - 7 Jul 2014). (Time series of NO$_2$ and HONO measurements were adapted from Tan et al, 2016)

[Figure]

**Figure R1.** Observed and simulated NO levels at the Wangdu site during the simulation period (27 Jun - 7 Jul 2014).

As illustrated, the original model underestimated the HONO by an order of magnitude during the campaign and was not able to predict ClNO$_2$ (was treated as an inert gas). With our new development, WRF-Chem with CBMZ_ReNOM significantly improved the performance in HONO, ClNO$_2$, as well as O$_3$ at the Wangdu site during the CareBeijing 2014 campaign. The CBMZ_ReNOM module well reproduced the level and the variation of N$_2$O$_5$ during the period of Jun 27-Jul 1, but overestimated the N$_2$O$_5$ concentration from Jul 2 to Jul 7. The overestimation of N$_2$O$_5$ is partly due to the underestimation in NO (Figure 2), which leads to

the underestimation of the $NO_3$ loss. The CBMZ_ReNOM module in general overestimated the concentration of the $ClNO_2$ during nighttime which is because of the overestimates in $N_2O_5$ during nighttime (Figure S3) and the possibly high uptake coefficient of $N_2O_5$. Compared with previous studies, our simulations in $N_2O_5$ and $ClNO_2$ were, overall, satisfactory.

We have revised our manuscript according to each specific comment from the reviewer and given point-by-point responses as bellow: (Our replies to the comments are highlighted in blue here, and all changes in the revised manuscript are highlighted in red)

**Specific comments**

**1.** pg 3 line 17 "To the best of our knowledge, no global or regional models, however, have simultaneously considered the sources/processes of HONO and ClNO2 and evaluated their regional impacts on the formation of O3 pollution in the boundary layer of the atmosphere." I am not sure the statement as written is true. Many studies have simultaneously considered HONO and ClNO2 as radical sources and showed how these species affect O3. For instance, Sarwar et al 2004 [GRL, 2014] studied O3 formation using CMAQV5.02 which contains the RACM2 mechanism [Goliff et al., AE 2013] and HONO chemistry. Also, Ahmadov et al. [ACP, 2015] and Edwards et al. [Nature, 2014] used models and HONO/ClNO2 data to investigate high wintertime ozone pollution events in an oil- and natural gas-producing region of the western US. There have also been numerous papers using 0D box models examining this chemistry.

**Response:** Although there have been some studies considering HONO *or* Cl chemistry in chemical transport models, they tend to investigate HONO or Cl chemistry (and their effects) separately (e.g. Sarwar 2004 and Goliff 2013 as the reviewer mentioned) and none of them has introduced the combined effects of these two reactive nitrogen species on ozone pollution at a regional scale. Besides, we developed the chemical module in a 3D regional chemical transport model and evaluated the impacts of these chemistry on ozone formations over China at a regional scale, which is different from 0-D box modelling studies.

**2.** pg 4 line 9 "reproduced the observed HONO by 85% on average" Not sure what this means.

**Response:** The sentence has been revised into 'We showed that including these additional sources of HONO very well simulated the observed HONO at a suburban site in southern China'.

**3.** pg 5 line 17. What is FMCl? A fluorine-metal compound?

**Response:** All definitions of the chemical species have been added in Table 1 in the revised manuscript. 'FMCl' means formyl chloride.

**4.** pg 7 line 7. "has been proved" A model cannot be proven, at least not in a mathematical sense. Why not simply say that this model has made reasonable predictions of PM2.5 and O3?

**Response:** Here we meant that the emission inventory was able to offer reasonable simulations. The sentence has been revised into "and this inventory has been suggested to offer reasonable model predictions of $PM_{2.5}$ and $O_3$ in multiple cities over China".

**5.** pg 7, section "2.2.3 O3 and NO2 measurement data" Please state how accurate these measurements are.

**Response:** These measurements in the China Ministry of Environmental Protection (MEP) air quality network have been conducted by each local environmental protection bureaus following the same standards for instrument operation and quality control set by the China MEP. The China MEP has set detailed technical specifications for installation, operation, and QA/QC for these stations which can found at http://english.sepa.gov.cn/Resources/standards/Air_Environment/(*in Chinese*). According to the standards, the accuracy for $O_3$ and NO2 measurement is ±5%.

**6.** pg 7, last line. The Mo converter also "detects" NO3, 2N2O5, HONO, ClNO2, PAN, and HNO3 to some degree as if it were NO2. The model should give some indication as what fraction of NOy is in the form of NOz (as a function of time of day); consider a sensitivity run in which the NO2 reported by the routine measurements is scaled down by this factor.

**Response:** As suggested, we used model simulated NOz to try to scale down the measured NO2* by using a factor: $NO_{2\ obs} = NO^*_{2\ obs} \times \frac{NO_{2\ mod}}{NO_{2\ mod} + NO_{z\ mod} - Nitrate_{mod}}$. (gas-phase HNO$_3$ is assumed to be converted into NO by 80% due to its loss on inlet (Xu et al., 2013); other gas-phase NOz species are assumed to be converted by 100% in the Mo converter) at each hour. Since uncertainties exist in the emission inventories, chemical models, and, thus, the final model results, this calculated scaling factor is certainly subject to a large uncertainty. Table S1 in the revised SI lists the statistics of the comparisons between the simulated and measured (both original and adjusted) NO$_2$. The simulations agreed well with the NO$_2$ measurements after our adjustment. Both the original and scaled NO$_2$ measurements were shown in Figure S1 in the supplementary materials.

**Table S1.** Statistics of model performance in the base and ReNOM cases for hourly NO$_2$ measurements (ppb) at the MEP air quality monitoring stations during the simulation period (27 Jun-7 Jul 2014).

| Region | Case | No. OBS | OBS[a] | MOD | COR | MB | RMSE | NMB | NME |
|---|---|---|---|---|---|---|---|---|---|

| Region | Case | No. OBS | OBS_scaled[a] | MOD | COR | MB | RMSE | NMB | NME |
|---|---|---|---|---|---|---|---|---|---|
| NCP | BASE | 48362 | 19.4 | 18.4 | 0.35 | -1.0 | 16.3 | -5.0% | 58.6% |
| NCP | ReNOM | | | 14.4 | 0.35 | -5.0 | 15.8 | -25.8% | 56.3% |
| YRD | BASE | 35421 | 17.7 | 27.0 | 0.21 | 9.3 | 23.5 | 52.3% | 96.9% |
| YRD | ReNOM | | | 19.2 | 0.17 | 1.4 | 17.9 | 8.1% | 75.8% |
| PRD | BASE | 15651 | 12.1 | 8.3 | 0.38 | -3.9 | 11.9 | -31.8% | 71.5% |
| PRD | ReNOM | | | 7.2 | 0.35 | -5.0 | 11.5 | -40.8% | 69.7% |
| China | BASE | 213308 | 15.6 | 14.2 | 0.32 | -1.4 | 16.3 | -9.2% | 73.1% |
| China | ReNOM | | | 10.9 | 0.32 | -4.8 | 14.6 | -30.5% | 67.4% |

| Region | Case | No. OBS | OBS_scaled[a] | MOD | COR | MB | RMSE | NMB | NME |
|---|---|---|---|---|---|---|---|---|---|
| NCP | BASE | 48362 | 15.6 | 18.4 | 0.48 | 2.8 | 14.4 | 17.9% | 62.2% |
| NCP | ReNOM | | 13.5 | 14.4 | 0.51 | 0.9 | 11.7 | 6.9% | 57.6% |
| YRD | BASE | 35421 | 14.4 | 27.0 | 0.37 | 12.6 | 23.3 | 87.3% | 113.9% |
| YRD | ReNOM | | 11.7 | 19.2 | 0.38 | 7.5 | 16.6 | 64.4% | 92.7% |
| PRD | BASE | 15651 | 10.0 | 8.3 | 0.37 | -1.7 | 11.0 | -17.1% | 77.0% |
| PRD | ReNOM | | 8.1 | 7.2 | 0.42 | -0.9 | 9.1 | -11.5% | 75.3% |
| China | BASE | 213308 | 12.3 | 14.2 | 0.43 | 1.9 | 14.7 | 15.2% | 77.6% |
| China | ReNOM | | 10.5 | 10.9 | 0.45 | 0.4 | 11.3 | 3.7% | 70.7% |

[a] OBS: original observations of $NO_2$; OBS_scaled: scaled observations of $NO_2$ based on model simulated reactive nitrogen species by using the equation of $NO_{2\ obs} = NO^*_{2\ obs} \times \frac{NO_{2\ mod}}{NO_{2\ mod} + NO_{Z\ mod} - Nitrate_{mod}}$, where $NO_2^*{}_{obs}$ is the original measurement of $NO_2$, $NO_{2\ mod}$ is the model simulation of $NO_2$, $NO_{Z\ mod}$ is the sum of simulations of HONO, $2 \times N_2O_5$, $ClNO_2$, $ClONO_2$, $NO_3$, $HNO_3$, $HNO_4$, PAN, and Nitrate, $Nitrate_{mod}$ is the simulated nitrate; gas-phase $HNO_3$ is assumed to be converted into NO by 80% in the Mo converter due to its possible loss on inlet; other gas-phase $NO_Z$ species are assumed to be converted by 100%.

**7.** pg 8 " 3.1.2 Spatial and vertical distributions of N2O5 and ClNO2 ". These predicted concentrations are considerably lower than recent measurements in the HK area (see sections further down).

**Response:** This study did not predict the elevated $N_2O_5$ levels in the HK-PRD region which is different from the results in Li et al. (2016), probably due to the different season (summer in this study compared to winter in the HK measurement) and hence the meteorological condition (southeasterly winds in summer and northerly winds in winter) and emission intensity (higher industrial emissions in winter than in summer). Besides, the measurements in HK were carried out at a mountain top site (~1000 m) and observations at such a high altitude was difficult to be resolved by the model with a resolution of 27 km. But our previous WRF-Chem simulations considering similar chlorine chemistry as this study and using a fine model resolution of 1 km gave satisfactory simulations of $N_2O_5$ and $ClNO_2$ at this mountain-top site in winter season (Li et al., 2016).

Besides, the uncertainties in emissions of NOx and chlorine, incomplete model parameterizations of formation and loss processes of $ClNO_2$ (e.g. Roberts et al., 2009) would

also influence our models results. A brief discussion has been added in the revised manuscript.

**8.** pg 9, line 11 "The model very well captured the measured HONO at Wangdu in the NCP region during a matching simulation period, reproducing 86% of the observations (0.81 vs. 0.94 ppb)" I don't understand this sentence. How does a model reproduce 86% of observations? Consider instead a scatter plot of model concentrations vs observations.

**Response:** The sentence has been revised into "The model very well captured the measured HONO at Wangdu in the NCP region with an average simulation of 0.81 ppb comparing with a mean observed value of 0.94 ppb". Detailed comparisons between the observations and simulations at Wangdu have been added in the supplementary materials.

9. pg 9, line 25 onward. Brown, S. S., et al. (2016), Nighttime chemistry at a high altitude site above Hong Kong, J. Geophys. Res.-Atmos., 121(5), 2457-2475, doi: 10.1002/2015jd024566 observed much higher concentrations than the model predicts. Please discuss.

**Response:** Measurements reported in Brown et al. (2016) were observed during a joint field campaign with our group in Hong Kong, as we cited (Wang et al., 2016) in Table 4. Similar to our response to the comment 7, the differences between model predictions and observations were probably due to a low model resolution that we applied and the differences between observation and simulation season. We have added the discussion in section 3.2 in the revised manuscript.

10. pg 18, Table 1. Please state the units of the reaction rates.

**Response:** Thanks a lot for the suggestion. This information has been added in Table 1.

11. rxn 7. Water should have a subscript.

**Response:** The subscript has been added.

12. rxn 26. What are PAR and X? Also, define the other terms, such as OLI, ALD2 etc.

**Response:** Thanks a lot for the suggestion. All definitions of the chemical species in Table 1 have been added.

**References**

Tan, Z., Fuchs, H., Lu, K., Hofzumahaus, A., Bohn, B., Broch, S., Dong, H., Gomm, S., Häseler, R.,

He, L., Holland, F., Li, X., Liu, Y., Lu, S., Rohrer, F., Shao, M., Wang, B., Wang, M., Wu, Y., Zeng, L., Zhang, Y., Wahner, A., and Zhang, Y.: Radical chemistry at a rural site (Wangdu) in the North China Plain: observation and model calculations of OH, HO2 and RO2 radicals, Atmos. Chem. Phys., 17, 663-690, 10.5194/acp-17-663-2017, 2017.

Tham, Y. J., Wang, Z., Li, Q., Yun, H., Wang, W., Wang, X., Xue, L., Lu, K., Ma, N., Bohn, B., Li, X., Kecorius, S., Größ, J., Shao, M., Wiedensohler, A., Zhang, Y., and Wang, T.: Significant concentrations of nitryl chloride sustained in the morning: investigations of the causes and impacts on ozone production in a polluted region of northern China, Atmos. Chem. Phys., 16, 14959-14977, 10.5194/acp-16-14959-2016, 2016.

Xu, Z., Wang, T., Xue, L. K., Louie, P. K. K., Luk, C. W. Y., Gao, J., Wang, S. L., Chai, F. H., and Wang, W. X.: Evaluating the uncertainties of thermal catalytic conversion in measuring atmospheric nitrogen dioxide at four differently polluted sites in China, Atmos. Environ., 76, 221-226, http://doi.org/10.1016/j.atmosenv.2012.09.043, 2013.